# Maternal smoking behaviour during pregnancy and the association of Sudden Unexpected Infant Death (SUID): A retrospective cohort study of births in the United States from 2017–2021

Kiki Hudson[1,2]*, Giulia M. Muraca[1,2]

**1** Department of Obstetrics and Gynecology, Faculty of Health Sciences, McMaster University, Hamilton, Ontario, Canada, **2** Department of Health Research Methods, Evidence, and Impact, McMaster University, Hamilton, Ontario, Canada

☙ These authors contributed equally to this work.
* kiki2@ualberta.ca

## Abstract

### Background

Maternal smoking during pregnancy is a significant risk factor for sudden unexpected infant death (SUID). However, the impact of variations in smoking behaviours, including timing, intensity, and cessation, remains understudied. This study examines the association between maternal smoking and SUID, incorporating detailed categorizations of smoking behaviours.

### Methods

We conducted a population-based, retrospective cohort study of live births in the United States from 2017 to 2021 using the Centre for Disease Control Linked Birth-Infant Death files. Maternal smoking was categorized as non-smoking, pre-pregnancy smoking only, trimester-specific smoking, continuous or discontinuous smoking, and cessation before the third trimester, with stratification by smoking intensity. SUID was defined using ICD-10 codes. Multivariable logistic regression was used to estimate unadjusted (OR) and adjusted odds ratios (aOR) for SUID. Sensitivity analyses examined mediation by gestational age and infant birth weight.

### Results

Heavy continuous smokers had the highest aOR for SUID (372.8 per 100,000 births; aOR 2.81, 95% CI 2.67–2.94), followed by light continuous smokers (395.6 per 100,000 births; aOR 2.47, 95% CI 2.19–2.78) and discontinuous heavy smokers (292.5 per 100,000 births; aOR 2.29, 95% CI 1.72–3.00) compared with non-smokers. Pre-pregnancy-only smokers had the lowest odds of SUID among all smoking categories (light: 188.2 per 100,000 births; aOR 1.77, 95% CI 1.48–2.10;

**Data availability statement:** All data are available at Centers for Disease Control and Prevention (CDC) specifically Period/Cohort Linked Birth-Infant Death Data Files. Accessed via URL: https://www.cdc.gov/nchs/data_access/vitalstatsonline.htm.

**Funding:** The author(s) received no specific funding for this work.

**Competing interests:** The authors have declared that no competing interests exist.

heavy: 152.8 per 100,000 births; aOR 1.61, 95% CI 1.44–1.78). In the sensitivity analysis, the natural indirect effect (NIE) of continuous smoking throughout pregnancy on SUID through gestational age and infant birth weight were insignificant (gestational age: β = 1.01, 95% CI 0.99–1.03, p = 0.28, infant birth weight: β = 1.04, 95% CI 0.99–1.08, p = 0.10).

## Conclusions

Maternal smoking significantly influences SUID, with earlier cessation exhibiting weaker associations. These findings emphasize the importance of early smoking cessation interventions to improve SUID outcomes.

## Introduction

Sudden Unexpected Infant Death (SUID) is the combination of primarily sleep-related deaths during infancy that are due to (1) sudden infant death syndrome (SIDS), (2) accidental suffocation or strangulation in bed (or any other location), and (3) ill-defined or other unknown causes [1,2]. It is the most common cause of infant mortality between one month and one year of age in high-income countries [1,3]. Recent estimates of SUID incidence in the United States (US) are between 90 and 100 deaths per 100,000 live births [4,5]. The most important risk factors for SUID are related to the sleep environment, with sleep position, bed-sharing with a parent, and soft sleeping surface being particularly critical [1]. Another important risk factor for SIDS is maternal smoke exposure [1]. This risk is most significant for maternal smoking while pregnant, but maternal smoking after birth and higher general tobacco exposure also increase the risk of SUID [6,7]. Studies have reported a fourfold greater risk of SUID among mothers who smoked during pregnancy compared to mothers who did not smoke [8]. Other known risk factors include young maternal age, no or poor prenatal care, prematurity, low infant birth weight, male sex, and maternal mental illness and substance abuse [1,9–14].

Over the last 20 years, the incidence of SUID has decreased by more than 50%. This has been largely credited to the efforts of the Back to Sleep campaign, a public health initiative promoting a safe infant sleeping environment [1]. However, SUID attributable to maternal smoking remains significant and preventable. In the US, approximately 3700 infants die annually from SUID, among which 22% can be attributed to smoking [15]. Previous studies characterizing the association between maternal smoking during pregnancy and SUID are limited by their definitions of maternal smoking, preventing assessment of the impact of distinct smoking behaviours, such as continuous smoking versus changes in smoking behaviours throughout pregnancy [15,16]. Furthermore, prior studies have not examined the impact of potential mediators such as prematurity and low infant birth weight on the association between maternal smoking and SUID. Lastly, several important confounders have not been accounted for in previous studies, including maternal obesity, pre-pregnancy hypertension, mother's nativity, and payment source for delivery

[17,18]. Accounting for these covariates is crucial because they capture important social, biological, and structural determinants of health that could confound or modify the observed association between maternal smoking and SUID. Therefore, to effectively address the ongoing issue of SUID, a more nuanced approach is needed.

Our objective was to assess the association between maternal smoking behaviour and SUID using seven categories of maternal smoking behaviour measurement: 1) no smoking, 2) smoking prior to pregnancy, 3) only smoked during the first trimester, 4) only smoked during the second or third trimester, 5) continuous smoking throughout pregnancy 6) discontinuous smoking throughout pregnancy, 7) smoking cessation during pregnancy. Additionally, we assessed the presence of a dose response in the relationship between maternal smoking and SUID through further categorization into heavy and light smokers for each maternal smoking behaviour.

## Methods

### Study design and data source

We used the Centers for Disease Control and Prevention (CDC) linked Birth-Infant Death Files to carry out a population-based cohort study of births in the US from 2017 to 2021 [19]. These data sets link infant death records with corresponding birth records, allowing for detailed analyses of maternal, pregnancy, and infant characteristics.

### Study population

The study included singleton, live births in the US occurring between January 1, 2017, and December 31, 2021, and infant deaths within the first year of life. Low birth weight is known to be highly associated with child mortality; thus, birth weights of less than 1000 grams were excluded to limit the effects of extremely low birth weights [20]. Furthermore, gestational ages of less than 25 weeks were excluded to limit extreme prematurity and to ensure all gestational ages reach at least one week into the third trimester. Infants with congenital anomalies were also excluded. Further exclusions were applied to missing data on exposure, outcome, or confounders and mediators, ensuring complete data for analysis (Fig 1).

### Exposure and outcome definitions

The primary exposure was maternal smoking behaviour, categorized into seven groups: 1) no pre-pregnancy and prenatal smoking, 2) only pre-pregnancy smoking, 3) only smoked during the first trimester, 4) only smoked during the second or third trimester, 5) continuous smoking (smoking during pre-pregnancy and throughout pregnancy or smoking throughout the three trimesters), 6) discontinuous smoking (stopped smoking during any period of time throughout pregnancy and started smoking again at a later time in their pregnancy), and 7) smoking cessation (continuous smoking during pre-pregnancy and through multiple trimesters and stopped before their third trimester). To calculate dose response, each of the smoking categories were further categorized into heavy and light smokers. Participants that smoked on average 20 cigarettes or more per day were considered heavy smokers and those that smoke on average less than 20 cigarettes per day were considered light smokers [21].

The outcome of interest was SUID, defined as the death of an infant under 365 days old, due to the following causes and corresponding ICD-10 codes, Sudden Infant Death Syndrome (R95), ill-defined and unknown causes (R99), or accidental suffocation and strangulation in bed (W75) [15].

### Covariates

Covariates included in our analysis were determined *a priori* based on previous studies [15–18]. Socioeconomic status (SES) variables including maternal ethnicity, marital status, maternal education, maternal nativity, Special Supplemental Nutrition Program for Women, Infants, and Children (WIC), and payment method. Maternal characteristics including maternal age, pre-pregnancy body mass index, pregnancy weight gain, pre-pregnancy hypertension, and gestational

 

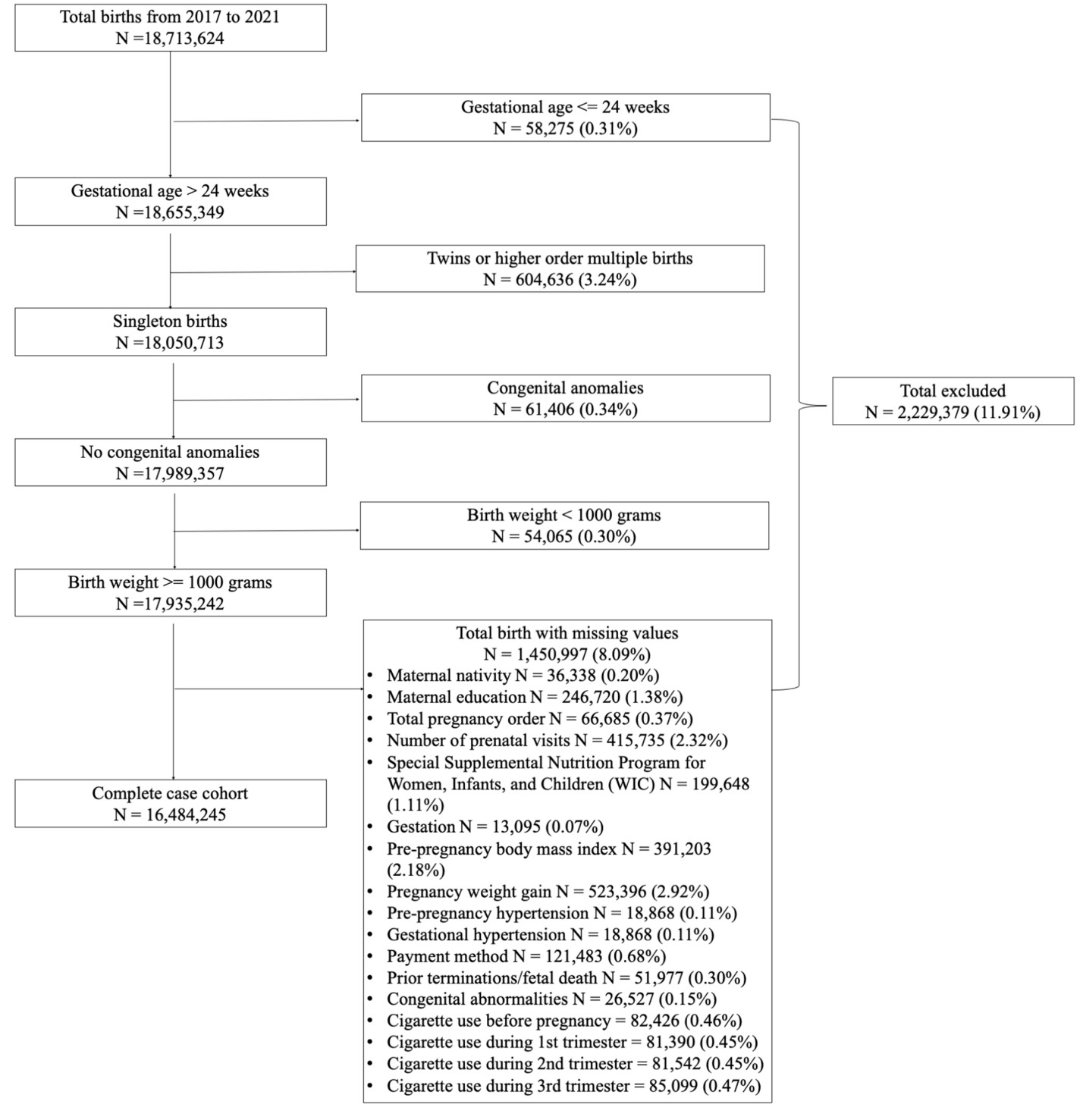

**Fig 1. Derivation of complete case cohort.**

hypertension. Other pregnancy characteristics including total pregnancy order, number of prenatal visits, and prior terminations or fetal death. Lastly, gestational age and infant birth weight have been identified as potential mediators based on their previously established associations with maternal smoking and SUID [22–24].

### Statistical analysis

Descriptive statistics on demographics and clinical characteristics of individuals with live births in the US were calculated for the study cohort and stratified by maternal smoking behaviour categories. We estimated crude SUID rates per 100,000 live births within each smoking behaviour category. Multivariable logistic regression models were used to estimate unadjusted and adjusted odds ratios (ORs) for the association between maternal smoking behaviour categories and SUID, comparing each maternal smoking behaviour to the no pre-pregnancy and prenatal smoking category, and controlling for previously determined confounders.

### Sensitivity analysis

Mediation analysis was carried out to investigate the roles of gestational age and infant birth weight as mediators in the relationship between maternal smoking behaviour and SUID. Gestational age and infant birth weight were categorized into binary variables. Gestational age under 37 weeks were categorized as preterm and infant birth weight under 2500 grams were categorized as "low" [24]. All mediation analysis examined the controlled direct effect (CDE; variable of interest effect on the outcome when the mediator is set to a fixed level), natural indirect effect (NIE; variable of interest effect on the outcome that is influenced by the mediator variable), natural direct effect (NDE; effect on the outcome after adjusting for the indirect effect exerted by the mediator), and total effect (overall variable of interest effect on the outcome, equal to NDE + NIE) [25].

## Results

There were 18,713,624 live births in the US between 2017 and 2021. After applying the exclusion criteria, the study population included 16,484,245 live births from 2017–2021 Demographics are outlined in detail in Table 1. Of the total population, 92% of patients were non-smokers (n = 15,222,196), 4% were heavy smokers who continuously smoked during their pregnancy (n = 664,741), 1.4% were heavy smokers who only smoked during pre-pregnancy (n = 234,259), and 0.94% were heavy smokers who ceased smoking before their third trimester (n = 154,618). The remaining maternal smoking behaviour groups (first trimester only heavy and light smokers, second or third trimester only heavy and light smokers, continuous light smokers, discontinuous heavy and light smokers, pre-pregnancy only light smokers, and smoking cessation light smokers) each represent less than 0.5% of the total population (Table 1).

Individuals who smoked, regardless of their smoking behaviour, had, on average, lower levels of education compared to non-smokers. Additionally, individuals who smoked had a higher prevalences of participation in the WIC program. Non-smokers had the lowest percentages of pre-pregnancy hypertension, infant gestational age under 37 weeks, and low infant birth weight (1000–2499 grams). Among all individuals who smoke, individuals who only smoked pre-pregnancy had the lowest rates of pre-pregnancy hypertension, infant gestational age under 37 weeks, and low infant birth weight (Table 1).

### Crude rates

In the overall population, the rate of SUID was 85.3 per 100,000 births. Compared with the rate of SUID among non-smokers (67.1 per 100,000 births), crude rates of SUID were higher in all maternal smoking behaviour groups, with light smokers who discontinuously smoked during their pregnancy (491.6 per 100,000 births) having the highest rates, followed by light continuous smokers (395.6 per 100,000 births), heavy continuous smokers (372.8 per 100,000 births), and

**Table 1. Demographic and maternal smoking behaviour of individuals with live births in the United States, 2017–2021.**

| | Maternal smoking behaviour throughout pregnancy | | | | | | | | | | | | | |
|---|---|---|---|---|---|---|---|---|---|---|---|---|---|---|
| | Overall N=16,484,245 | Nonsmoker N=15,222,196 | 1st Trimester Light N=1,980 | Pre-pregnancy Only Heavy N=234,259 | Pre-pregnancy Only Light N=69,620 | Smoking Cessation Heavy N=154,618 | 2nd or 3rd Trimester Heavy N=4,406 | Smoking Cessation Light N=33,362 | Discontinuous Smoking Heavy N=17,096 | 1st Trimester Heavy N=3,692 | Continuous Smoking Heavy N=664,741 | Continuous Smoking Light N=71,790 | Discontinuous Smoking Light N=4,067 | 2nd or 3rd Trimester Light N=2,418 |
| **SUID** | | | | | | | | | | | | | | |
| No | 16,470,181 (100%) | 15,211,987 (100%) | 1,977 (100%) | 233,901 (100%) | 69,489 (100%) | 154,217 (100%) | 4,394 (100%) | 33,269 (100%) | 17,046 (100%) | 3,679 (100%) | 662,263 (100%) | 71,506 (100%) | 4,047 (100%) | 2,406 (100%) |
| Yes | 14,064 (<0.1%) | 10,209 (<0.1%) | 3 (0.2%) | 358 (0.2%) | 131 (0.2%) | 401 (0.3%) | 12 (0.3%) | 93 (0.3%) | 50 (0.3%) | 13 (0.4%) | 2,478 (0.4%) | 284 (0.4%) | 20 (0.5%) | 12 (0.5%) |
| **Maternal Race** | | | | | | | | | | | | | | |
| White | 12,211,626 (74%) | 11,192,488 (74%) | 1,182 (60%) | 190,557 (81%) | 49,051 (70%) | 124,881 (81%) | 3,215 (73%) | 21,079 (63%) | 13,802 (81%) | 2,648 (72%) | 562,796 (85%) | 45,921 (64%) | 2,567 (63%) | 1,439 (60%) |
| Black | 2,546,115 (15%) | 2,393,199 (16%) | 583 (29%) | 26,628 (11%) | 13,123 (19%) | 17,945 (12%) | 812 (18%) | 8,418 (25%) | 1,870 (11%) | 703 (19%) | 61,943 (9.3%) | 19,190 (27%) | 1,000 (25%) | 701 (29%) |
| American Indian and Alaska Native | 155,263 (0.9%) | 130,869 (0.9%) | 75 (3.8%) | 3,508 (1.5%) | 2,168 (3.1%) | 2,828 (1.8%) | 122 (2.8%) | 1,304 (3.9%) | 392 (2.3%) | 102 (2.8%) | 11,184 (1.7%) | 2,394 (3.3%) | 203 (5.0%) | 114 (4.7%) |
| Asian | 1,077,222 (6.5%) | 1,068,316 (7.0%) | 39 (2.0%) | 2,689 (1.1%) | 1,451 (2.1%) | 1,166 (0.8%) | 41 (0.9%) | 461 (1.4%) | 70 (0.4%) | 73 (2.0%) | 2,182 (0.3%) | 678 (0.9%) | 28 (0.7%) | 28 (1.2%) |
| Native Hawaiian or Other Pacific Islander | 50,363 (0.3%) | 48,025 (0.3%) | 5 (0.3%) | 507 (0.2%) | 237 (0.3%) | 309 (0.2%) | 18 (0.4%) | 104 (0.3%) | 34 (0.2%) | 11 (0.3%) | 906 (0.1%) | 182 (0.3%) | 18 (0.4%) | 7 (0.3%) |
| More than one | 443,656 (2.7%) | 389,299 (2.6%) | 96 (4.8%) | 10,370 (4.4%) | 3,590 (5.2%) | 7,489 (4.8%) | 198 (4.5%) | 1,996 (6.0%) | 928 (5.4%) | 155 (4.2%) | 25,730 (3.9%) | 3,425 (4.8%) | 251 (6.2%) | 129 (5.3%) |
| **Maternal Education** | | | | | | | | | | | | | | |
| 8th grade or less | 499,811 (3.0%) | 480,184 (3.2%) | 54 (2.7%) | 1,659 (0.7%) | 733 (1.1%) | 1,444 (0.9%) | 104 (2.4%) | 426 (1.3%) | 238 (1.4%) | 62 (1.7%) | 13,332 (2.0%) | 1,455 (2.0%) | 74 (1.8%) | 46 (1.9%) |
| 9th through 12th grade with no diploma | 1,481,857 (9.0%) | 1,234,827 (8.1%) | 428 (22%) | 27,412 (12%) | 8,974 (13%) | 24,261 (16%) | 1,110 (25%) | 6,211 (19%) | 3,480 (20%) | 767 (21%) | 155,398 (23%) | 17,504 (24%) | 854 (21%) | 631 (26%) |
| High school graduate or GED completed | 4,269,930 (26%) | 3,735,290 (25%) | 814 (41%) | 90,326 (39%) | 24,700 (35%) | 64,714 (42%) | 1,883 (43%) | 13,583 (41%) | 7,383 (43%) | 1,598 (43%) | 295,909 (45%) | 31,044 (43%) | 1,705 (42%) | 981 (41%) |
| Some college credit, but not a degree | 3,238,307 (20%) | 2,923,977 (19%) | 471 (24%) | 69,550 (30%) | 20,186 (29%) | 44,178 (29%) | 896 (20%) | 8,969 (27%) | 4,367 (26%) | 855 (23%) | 147,854 (22%) | 15,463 (22%) | 998 (25%) | 543 (22%) |

*(Continued)*

| | Overall N=16,484,245 | Nonsmoker N=15,222,196 | 1st Trimester Light N=1,980 | Pre-pregnancy Only Heavy N=234,259 | Pre-pregnancy Only Light N=69,620 | Smoking Cessation Heavy N=154,618 | 2nd or 3rd Trimester Heavy N=4,406 | Smoking Cessation Light N=33,362 | Discontinuous Smoking Heavy N=17,096 | 1st Trimester Heavy N=3,692 | Continuous Smoking Heavy N=664,741 | Continuous Smoking Light N=71,790 | Discontinuous Smoking Light N=4,067 | 2nd or 3rd Trimester Light N=2,418 |
|---|---|---|---|---|---|---|---|---|---|---|---|---|---|---|
| **Maternal smoking behaviour throughout pregnancy** | | | | | | | | | | | | | | |
| Associate degree | 1,390,978 (8.4%) | 1,310,419 (8.6%) | 118 (6.0%) | 20,955 (8.9%) | 5,701 (8.2%) | 11,228 (7.3%) | 259 (5.9%) | 2,204 (6.6%) | 1,040 (6.1%) | 229 (6.2%) | 34,646 (5.2%) | 3,796 (5.3%) | 255 (6.3%) | 128 (5.3%) |
| Bachelor's degree | 3,502,385 (21%) | 3,449,926 (23%) | 74 (3.7%) | 18,965 (8.1%) | 6,994 (10%) | 7,163 (4.6%) | 116 (2.6%) | 1,588 (4.8%) | 484 (2.8%) | 138 (3.7%) | 14,697 (2.2%) | 2,039 (2.8%) | 140 (3.4%) | 61 (2.5%) |
| Master's degree | 1,625,905 (9.9%) | 1,614,376 (11%) | 17 (0.9%) | 4,689 (2.0%) | 1,973 (2.8%) | 1,408 (0.9%) | 26 (0.6%) | 319 (1.0%) | 87 (0.5%) | 33 (0.9%) | 2,498 (0.4%) | 422 (0.6%) | 34 (0.8%) | 23 (1.0%) |
| Doctorate or Professional Degree | 475,072 (2.9%) | 473,197 (3.1%) | 4 (0.2%) | 703 (0.3%) | 359 (0.5%) | 222 (0.1%) | 12 (0.3%) | 62 (0.2%) | 17 (<0.1%) | 10 (0.3%) | 407 (<0.1%) | 67 (<0.1%) | 7 (0.2%) | 5 (0.2%) |
| **WIC** | | | | | | | | | | | | | | |
| No | 10,903,059 (66%) | 10,317,631 (68%) | 875 (44%) | 129,303 (55%) | 37,508 (54%) | 73,756 (48%) | 1,926 (44%) | 15,001 (45%) | 7,152 (42%) | 1,698 (46%) | 283,716 (43%) | 31,695 (44%) | 1,729 (43%) | 1,069 (44%) |
| Yes | 5,581,186 (34%) | 4,904,565 (32%) | 1,105 (56%) | 104,956 (45%) | 32,112 (46%) | 80,862 (52%) | 2,480 (56%) | 18,361 (55%) | 9,944 (58%) | 1,994 (54%) | 381,025 (57%) | 40,095 (56%) | 2,338 (57%) | 1,349 (56%) |
| **Number of Prenatal Visits** | | | | | | | | | | | | | | |
| 0-6 visits | 1,629,431 (9.9%) | 1,408,561 (9.3%) | 380 (19%) | 20,174 (8.6%) | 7,781 (11%) | 19,731 (13%) | 1,036 (24%) | 5,877 (18%) | 2,857 (17%) | 559 (15%) | 143,300 (22%) | 17,762 (25%) | 787 (19%) | 626 (26%) |
| 7-10 visits | 4,969,577 (30%) | 4,593,786 (30%) | 635 (32%) | 64,082 (27%) | 20,749 (30%) | 45,784 (30%) | 1,366 (31%) | 10,539 (32%) | 5,203 (30%) | 1,128 (31%) | 201,847 (30%) | 22,397 (31%) | 1,285 (32%) | 776 (32%) |
| 11-14 visits | 7,344,295 (45%) | 6,858,077 (45%) | 694 (35%) | 107,983 (46%) | 30,130 (43%) | 64,467 (42%) | 1,447 (33%) | 12,626 (38%) | 6,678 (39%) | 1,428 (39%) | 235,044 (35%) | 23,479 (33%) | 1,512 (37%) | 730 (30%) |
| 15 visits and over | 2,540,942 (15%) | 2,361,772 (16%) | 271 (14%) | 42,020 (18%) | 10,960 (16%) | 24,636 (16%) | 557 (13%) | 4,320 (13%) | 2,358 (14%) | 577 (16%) | 84,550 (13%) | 8,152 (11%) | 483 (12%) | 286 (12%) |
| **Gestation** | | | | | | | | | | | | | | |
| under 37 weeks | 1,574,357 (9.6%) | 1,412,529 (9.3%) | 284 (14%) | 22,035 (9.4%) | 6,696 (9.6%) | 18,026 (12%) | 647 (15%) | 4,185 (13%) | 1,804 (11%) | 435 (12%) | 95,756 (14%) | 11,120 (15%) | 479 (12%) | 361 (15%) |
| 37-38 weeks | 4,322,595 (26%) | 3,979,105 (26%) | 497 (25%) | 59,687 (25%) | 17,707 (25%) | 39,504 (26%) | 1,235 (28%) | 8,830 (26%) | 4,691 (27%) | 936 (25%) | 188,379 (28%) | 20,284 (28%) | 1,086 (27%) | 654 (27%) |
| 39-40 weeks | 8,433,504 (51%) | 7,849,221 (52%) | 915 (46%) | 118,393 (51%) | 34,875 (50%) | 74,128 (48%) | 1,933 (44%) | 15,490 (46%) | 8,096 (47%) | 1,787 (48%) | 294,504 (44%) | 31,211 (43%) | 1,912 (47%) | 1,039 (43%) |
| 41 weeks and over | 2,153,789 (13%) | 1,981,341 (13%) | 284 (14%) | 34,144 (15%) | 10,342 (15%) | 22,960 (15%) | 591 (13%) | 4,857 (15%) | 2,505 (15%) | 534 (14%) | 86,102 (13%) | 9,175 (13%) | 590 (15%) | 364 (15%) |
| **Infant Birth Weight** | | | | | | | | | | | | | | |
| 1000-2499 grams | 972,814 (5.9%) | 844,796 (5.5%) | 183 (9.2%) | 13,923 (5.9%) | 4,410 (6.3%) | 12,940 (8.4%) | 480 (11%) | 3,128 (9.4%) | 1,365 (8.0%) | 295 (8.0%) | 81,687 (12%) | 8,961 (12%) | 371 (9.1%) | 275 (11%) |

*(Continued)*

**Table 1.** (Continued)

| | Overall N=16,484,245 | Nonsmoker N=15,222,196 | 1st Trimester Light N=1,980 | Pre-pregnancy Only Heavy N=234,259 | Pre-pregnancy Only Light N=69,620 | Smoking Cessation Heavy N=154,618 | 2nd or 3rd Trimester Heavy N=4,406 | Smoking Cessation Light N=33,362 | Discontinuous Smoking Heavy N=17,096 | 1st Trimester Heavy N=3,692 | Continuous Smoking Heavy N=664,741 | Continuous Smoking Light N=71,790 | Discontinuous Smoking Light N=4,067 | 2nd or 3rd Trimester Light N=2,418 |
|---|---|---|---|---|---|---|---|---|---|---|---|---|---|---|
| **Maternal smoking behaviour throughout pregnancy** | | | | | | | | | | | | | | |
| 2500-2999 grams | 3,049,401 (18%) | 2,737,600 (18%) | 457 (23%) | 41,941 (18%) | 13,371 (19%) | 32,043 (21%) | 1,184 (27%) | 7,561 (23%) | 4,016 (23%) | 813 (22%) | 189,078 (28%) | 19,683 (27%) | 1,051 (26%) | 603 (25%) |
| 3000-3499 grams | 6,628,254 (40%) | 6,139,260 (40%) | 792 (40%) | 91,468 (39%) | 27,707 (40%) | 59,851 (39%) | 1,773 (40%) | 13,038 (39%) | 7,025 (41%) | 1,409 (38%) | 256,129 (39%) | 27,238 (38%) | 1,627 (40%) | 937 (39%) |
| 3500-3999 grams | 4,522,233 (27%) | 4,257,048 (28%) | 446 (23%) | 66,114 (28%) | 18,721 (27%) | 38,451 (25%) | 804 (18%) | 7,526 (23%) | 3,731 (22%) | 935 (25%) | 114,105 (17%) | 13,052 (18%) | 824 (20%) | 476 (20%) |
| 4000 grams and over | 1,311,543 (8.0%) | 1,243,492 (8.2%) | 102 (5.2%) | 20,813 (8.9%) | 5,411 (7.8%) | 11,333 (7.3%) | 165 (3.7%) | 2,109 (6.3%) | 959 (5.6%) | 240 (6.5%) | 23,742 (3.6%) | 2,856 (4.0%) | 194 (4.8%) | 127 (5.3%) |

individuals who only smoked during their second or third trimesters (351.7 per 100,000 births). Similar rates of SUID were observed among heavy smokers who discontinuously smoked during their pregnancy (292.5 per 100,000 births), individuals who only smoked during their first trimester (282.1 per 100,000 births), and light and heavy smokers who ceased smoking before their third trimester (light: 278.8 per 100,000 births; heavy: 259.35 per 100,000 births). Light and heavy smokers who only smoked during pre-pregnancy had the lowest rates of SUID (light: 188.2 per 100,000 births; heavy: 152.8 per 100,000 births) among all individuals that smoked (Table 2). Power calculations indicated that heavy and light categories for only smoking during the first trimester and only smoking during the second or third trimesters had insufficient sample sizes. Thus, heavy and light categories were combined, and dose responses were not examined for these maternal smoking behaviour categories.

## Adjusted results

All maternal smoking behaviour categories were compared to non-smokers (67.1 per 100,000 births). After adjusting for SES, maternal characteristics, and pregnancy characteristics, light smokers who discontinuously smoked during their pregnancy had the highest adjusted odds of SUID (491.6 per 100,000 births; aOR 3.27, 95% CI 2.03–4.93). They are followed by continuous heavy (372.8 per 100,000 births; aOR 2.81, 95% CI 2.67–2.94) and light smokers (395.6 per 100,000 births; aOR 2.47, 95% CI 2.19–2.78). Individuals who only smoked during their second or third trimesters (351.70 per 100,000 births; aOR 2.30, 95% CI 1.50–3.36) had similar odds of SUID with heavy smokers who smoked discontinuously throughout their pregnancy (292.5 per 100,000 births; aOR 2.29, 95% CI 1.72–3.00). However, higher odds of SUID were observed for individuals who only smoked during their second or third trimesters (351.70 per 100,000 births; aOR 2.30, 95% CI 1.50–3.36) compared to individuals who only smoked during their first trimester (282.09 per 100,000 births; aOR 2.14, 95% CI 1.25–3.38). Lastly, individuals who only smoked during pre-pregnancy had the lowest adjusted odds of SUID (light: 188.2 per 100,000 births; aOR 1.77, 95% CI 1.48–2.10; heavy: 152.8 per 100,000 births; aOR 1.61, 95% CI 1.44–1.78) (Table 2).

Among maternal smoking behaviour categories that were accessed for dose response, higher odds of SUID were observed among heavy smokers in individuals who continuously smoked (heavy: aOR 2.81, 95% CI 2.67–2.94; light: aOR 2.47, 95% CI 2.19–2.78) and among individuals who ceased smoking before their third trimester (heavy: aOR 2.30, 95% CI 2.07–2.54; light: aOR 2.00, 95% CI 1.62–2.44). For individuals that discontinuously smoked throughout their pregnancy (light: aOR 3.27, 95% CI 1.72–3.00; heavy: aOR 2.29, 95% CI 1.72–3.00) and for those who only smoked during

**Table 2. Unadjusted and adjusted associations between sudden unexpected infant death (SUID) by maternal smoking behaviour categories compared to non-smoking category, United States, 2017-2021.**

| Maternal smoking behaviour categories | n/N | Rate per 100,000 | OR | 95% CI | aOR | 95% CI |
|---|---|---|---|---|---|---|
| Nonsmoking | 10209/15222196 | 67.1 | Ref | Ref | Ref | Ref |
| Continuous smoking (heavy) | 2478/664741 | 372.8 | 5.58 | 5.33–5.82 | 2.81 | 2.67–2.94 |
| Continuous smoking (light) | 284/71790 | 395.6 | 5.92 | 5.25–6.65 | 2.47 | 2.19–2.78 |
| Pre-pregnancy only (heavy) | 358/234259 | 152.8 | 2.28 | 2.05–2.53 | 1.61 | 1.44–1.78 |
| Pre-pregnancy only (light) | 131/69620 | 188.2 | 2.81 | 2.35–3.32 | 1.77 | 1.48–2.10 |
| Only 1st trimester smoking | 16/5672 | 282.1 | 4.22 | 2.47–6.64 | 2.14 | 1.25–3.38 |
| Only 2nd or 3rd trimester smoking | 24/6824 | 351.7 | 5.26 | 3.42–7.66 | 2.30 | 1.50–3.36 |
| Discontinuous smoking (heavy) | 50/17096 | 292.5 | 4.37 | 3.27–5.70 | 2.29 | 1.72–3.00 |
| Discontinuous smoking (light) | 20/4067 | 491.6 | 7.36 | 4.58–11.10 | 3.27 | 2.03–4.93 |
| Smoking cessation (heavy) | 401/154618 | 259.4 | 3.87 | 3.50–4.28 | 2.30 | 2.07–2.54 |
| Smoking cessation (light) | 93/33362 | 278.8 | 4.17 | 3.37–5.08 | 2.00 | 1.62–2.44 |

OR, odds ratio; aOR, adjusted odds ratio; CI, confidence interval

pre-pregnancy (light: aOR 1.77, 95% CI 1.48–2.10; heavy: aOR 1.61, 95% CI 1.44–1.78), higher odds of SUID were observed among light smokers. However, CI overlapped for heavy and light smokers in every maternal smoking behaviour category (Fig 2).

### Sensitivity analysis

Due to computation limitations, mediation analyses on birth weight and gestational age were only conducted for the continuous smoking category using a randomly generated subset of 500,000 samples. Furthermore, due to conversion issues, models used for mediation analyses needed to reduce model dimensionality. Therefore, maternal education, maternal age, number of prenatal visits, and payment method were excluded from the mediation model.

For gestational age, the OR for the NIE of continuous smoking throughout pregnancy on SUID through gestational age was insignificant (β = 1.01, 95% CI 0.99–1.03, p = 0.28). OR for NDE of continuous smoking throughout pregnancy on SUID was β = 2.34 (95% CI 1.68–3.01, p < 0.01), and OR for CDE was β = 2.35 (95% CI 1.62–3.08, p < 0.01). OR for the total effect of continuous smoking throughout pregnancy on SUID was β = 2.37 (95% CI 1.70–3.03, p < 0.01), with 1.71% (95% CI −1.44–4.85, p = 0.29) of the effect mediated by gestational age.

For infant birth weight, OR for NIE of continuous smoking throughout pregnancy on SUID through infant birth weight was insignificant (β = 1.04, 95% CI 0.99–1.08, p = 0.10). OR for NDE was β = 2.25 (95% CI 1.59–2.90, p < 0.01) and OR for CDE was β = 2.24 (95% CI 2.94–3.48, p < 0.01). The OR for the total effect of continuous smoking throughout pregnancy on SUID was β = 2.32 (95% CI 1.67–2.99, p < 0.01), with 6.28% (95% CI −1.30–13.87, p = 0.10) of the effect mediated by infant birth weight.

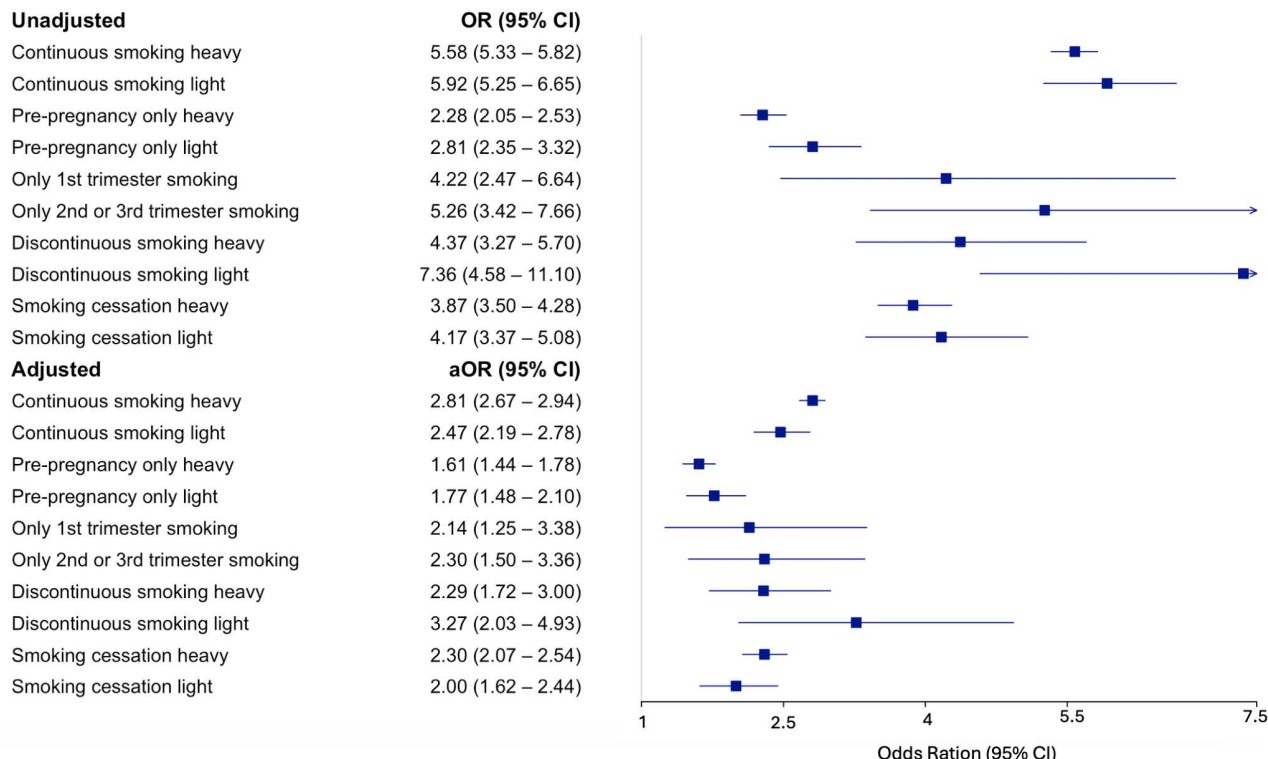

**Fig 2. Forrest plot of unadjusted and adjusted odds ratios of association between SUID by maternal smoking behaviour categories compared to non-smoking category, United States, 2017-2021.**

## Discussion

This population-based, retrospective cohort study of live births in the US from 2017 to 2021 revealed significant associations between maternal smoking behaviour and SUID. After adjusting for these socio-economic disparities as well as key pregnancy and infant factors, we observed the highest SUID rates among continuous smokers (heavy and light). While discontinuous light smokers appeared to have the highest point estimate, the wide confidence interval suggests greater uncertainty around this estimate. Importantly, the confidence interval overlaps with all other smoking behaviour categories, indicating that the apparent difference between these groups may not be statistically meaningful. Additionally, we saw a trend suggesting higher odds of SUID among individuals smoking later in pregnancy. Sensitivity analyses found minimal mediation by gestational age and birth weight. These findings support our hypothesis that all smoking behaviours would have increased odds of SUID, with continuous smoking having the highest. However, no differences were observed between heavy and light smokers for any category.

Our findings align with prior studies that consistently demonstrate any maternal smoking during pregnancy increases the risk of SUID [12,13]. This is consistent with existing theoretical positions emphasizing smoking's detrimental impact on fetal and infant health. However, unlike some previous studies that observed a clear dose-response relationship across all smoking categories, our study did not find significant dose-response associations in any smoking behaviours. This discrepancy may stem from our use of categorical analysis, which may lack the granularity of linear modelling approaches traditionally used in this context. Misclassification is also a concern, as "light" smokers in our study may include individuals with smoking habits near the threshold of 20 cigarettes per day, potentially diluting dose-response patterns. Unexpected results, such as higher odds of SUID among light smokers in certain categories, may also reflect unmeasured variables. For instance, the lack of detailed data on smoking behaviours prior to pregnancy limited our ability to fully account for cumulative exposure. The omission of detailed data on smoking behaviours prior to pregnancy could lead to an underestimation or overestimation of the true association between smoking and SUID, as the model may not accurately capture the cumulative effects of nicotine exposure.

Our results reinforce the need for public health interventions targeting maternal smoking cessation, especially early in pregnancy or before conception. By demonstrating that even intermittent or low intensity smoking increases SUID risk, our study highlights the importance of framing any smoking during pregnancy as harmful. Based on our findings, smoking cessation prior to pregnancy could reduce the risk of SUID by 20%. Thus, early interventions, such as integrating smoking cessation counselling into prenatal care or WIC programs, could have significant public health benefits [26,27]. For instance, leveraging WIC participation as an opportunity for smoking cessation education could help address the socio-economic disparities linked to smoking behaviours [28]. Furthermore, these findings highlight the need to reframe public health messaging around smoking during pregnancy. Educational campaigns should emphasize that cessation at any point reduces risk, but quitting earlier offers the most protective benefits. Policymakers could enhance access to smoking cessation resources, such as subsidized nicotine replacement therapies, especially for high-risk populations [29,30]. These findings also challenge healthcare providers and policymakers to address socioeconomic and behavioural factors that sustain smoking during pregnancy, implementing multi-faceted strategies to address SUID.

This study has several strengths. First, by differentiating between light and heavy smoking, continuous and discontinuous smoking, and timing of smoking during pregnancy, the study captured important variations in exposure. Additionally, this study adjusted for a comprehensive range of confounders, including socioeconomic factors, maternal health conditions, and pregnancy characteristics. Lastly, sensitivity analyses examined gestational age and infant birth weight as potential mediators, which provided valuable insights into the pathways linking maternal smoking to SUID. However, several limitations still exist. First, the study lacked data on maternal smoking behaviours prior to pregnancy, preventing the examination of cumulative tobacco exposure and its long-term effects on SUID. Additionally, data on alcohol or other substance use, which may interact with smoking, were unavailable. Lastly, sensitivity analyses were limited to a subset of continuously smoking individuals due to computation constraints, and non-bootstrap confidence intervals may affect the

robustness of some estimates. Although the study made significant efforts to mitigate potential biases through confounder adjustment, the potential for residual confounding remains, particularly given the complex interplay of factors influencing SUID.

Despite these limitations, the study provides a valuable contribution to the literature by utilizing large-scale, nationally representative data and employing detailed categorizations of smoking behaviours. The findings from this study open several avenues for future research to better understand and address the relationship between maternal smoking and SUID. Future studies should examine the long-term impact of maternal smoking habits prior to pregnancy and the impact of changes in these habits during pregnancy on SUID outcomes. Additionally, studies should incorporate data on alcohol and other substance use, as well as potential interactions between these factors and smoking, to provide a more comprehensive understanding of their combined effects. Finally, investigating the complex interplay of socioeconomic, environmental, and biological factors through interdisciplinary approaches may help address residual confounding and uncover novel pathways influencing SUID risk. These efforts would strengthen the evidence base and inform targeted interventions to reduce SUID incidence.

## Supporting information

**S1 Appendix. DAG of maternal smoking behaviour and SUID.**
(DOCX)

**S2 Table. Additional Demographic and maternal smoking behaviours of individuals with live births in the United States, 2017–2021.**
(DOCX)

**S3 Appendix. Smoking trend among individuals with live birth in the U.S. from 2017 to 2021.**
(DOCX)

**S4 Appendix. SUID trend among individuals with live birth in the U.S. from 2017 to 2021.**
(DOCX)

**S5 Appendix. STROBE Statement Checklist of items that should be included in reports of cohort studies.**
(DOC)

## Acknowledgments

We want to acknowledge the HRM 760: Applied Epidemiological Methods in Secondary Analysis course at McMaster University for the inception of this project, and the Perinatal Epidemiological Research Lab at McMaster University for its continual support.

## Author contributions

**Conceptualization:** Kiki Hudson, Giulia M. Muraca.

**Data curation:** Kiki Hudson.

**Formal analysis:** Kiki Hudson.

**Funding acquisition:** Giulia M. Muraca.

**Investigation:** Kiki Hudson, Giulia M. Muraca.

**Methodology:** Kiki Hudson.

**Project administration:** Giulia M. Muraca.

**Software:** Kiki Hudson.

**Supervision:** Giulia M. Muraca.

**Validation:** Giulia M. Muraca.

**Visualization:** Kiki Hudson.

**Writing – original draft:** Kiki Hudson.

**Writing – review & editing:** Kiki Hudson, Giulia M. Muraca.

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
