## [Decision Letter · Decision Letter 0]

15 Dec 2025

PONE-D-25-50358Maternal Smoking Behaviour During Pregnancy and the Association of Sudden Unexpected Infant Death (SUID): A Cohort StudyPLOS One

Dear Dr. Hudson,

Thank you for submitting your manuscript to PLOS ONE. After careful consideration, we feel that it has merit but does not fully meet PLOS ONE’s publication criteria as it currently stands. Therefore, we invite you to submit a revised version of the manuscript that addresses the points raised during the review process.

We look forward to receiving your revised manuscript.

Kind regards,

Malshani Lakshika Pathirathna, PhD

Academic Editor

PLOS One

2. Please update your submission to use the PLOS LaTeX template. The template and more information on our requirements for LaTeX submissions can be found at http://journals.plos.org/plosone/s/latex

Additional Editor Comments (if provided):

Reviewers' comments:

Reviewer's Responses to Questions

**Comments to the Author**

1. Is the manuscript technically sound, and do the data support the conclusions?

Reviewer #1: Yes

Reviewer #2: Yes

2. Has the statistical analysis been performed appropriately and rigorously? 

Reviewer #1: Yes

Reviewer #2: Yes

3. Have the authors made all data underlying the findings in their manuscript fully available?

Reviewer #1: Yes

Reviewer #2: Yes

4. Is the manuscript presented in an intelligible fashion and written in standard English?

Reviewer #1: Yes

Reviewer #2: Yes

5. Review Comments to the Author

Reviewer #1: The work presented in this manuscript is important and it is very well written. However, the title does not indicate the study setting and no indication of the type of study. Other than that, I do not believe that any editing is required except, may be in formatting and removing the title, sub title levels given within parenthesis and the numbering and titles of figures.

Reviewer #2: This is a well-written clear manuscript that adds to the literature by incorporating 'heaviness' and 'timing' of smoking during pregnancy into the SUID literature. The only issues I have with the paper are 1. there IS a paper examining the relationship between maternal obesity and SUID (https://pubmed.ncbi.nlm.nih.gov/39073792/)...Seemingly, obesity (which is available in the CDC dataset) should be a covariate. and 2. the authors should re-order table 2, from highest to lowest aOR, and then describe findings in the results section (and abstract) going down from higherst to lowest. As it is, the jumping around makes it confusing to a reader. Similarly, reorient the forest plot in the same way.

6. PLOS authors have the option to publish the peer review history of their article (what does this mean? ). If published, this will include your full peer review and any attached files.). If published, this will include your full peer review and any attached files.

**Do you want your identity to be public for this peer review?** For information about this choice, including consent withdrawal, please see our For information about this choice, including consent withdrawal, please see our Privacy Policy ..

Reviewer #1: No

Reviewer #2: **Yes:** William B Weeks, MD, PhD, MBAWilliam B Weeks, MD, PhD, MBA

---

## [Author Response · Author response to Decision Letter 1]

25 Jan 2026

Reviewer #1

Comment 1. The work presented in this manuscript is important and it is very well written. However, the title does not indicate the study setting and no indication of the type of study. Other than that, I do not believe that any editing is required except, may be in formatting and removing the title, sub title levels given within parenthesis and the numbering and titles of figures.

Response: Thank you. We have indicated the study setting and study type into the long title, changing it to “Maternal Smoking Behaviour During Pregnancy and the Association of Sudden Unexpected Infant Death (SUID): A Retrospective Cohort Study of Births in the United States from 2017-2021”. We have also changed the heading levels formats, and added an acknowledgement section (lines 301-304) and supporting information section (lines 406-414). Lastly, we referred to the PLOS One submission guide to ensure that our figures, table numbers, titles, and citations fit the criteria.

Reviewer #2

Comment 1. This is a well-written clear manuscript that adds to the literature by incorporating 'heaviness' and 'timing' of smoking during pregnancy into the SUID literature.

Response: Thank you.

Comment 2. There IS a paper examining the relationship between maternal obesity and SUID (https://pubmed.ncbi.nlm.nih.gov/39073792/)...Seemingly, obesity (which is available in the CDC dataset) should be a covariate.

Response: Thank you for pointing out the lack of maternal obesity indicated as covariate despite it being included in our analysis. We have indicated pregnancy body mass index as a covariate in our manuscript (lines 121), have added its distribution into the S2 Table, and added reference #18: Tanner D, et al. Maternal obesity and risk of sudden unexpected infant death. JAMA Pediatrics. 2024;178(9):906-913. doi: 10.1001/jamapediatrics.2024.2455.

Comment 3. The authors should re-order table 2, from highest to lowest aOR, and then describe findings in the results section (and abstract) going down from highest to lowest. As it is, the jumping around makes it confusing to a reader. Similarly, reorient the forest plot in the same way.

Response: We have reordered our findings in Crude Rate (line 168-177) and Adjusted Results (line 191-199, line 204-207 going from highest to lowest. However, regarding the forest plot and Table 2, while we understand the reviewer’s point, given that our study seeks to examine duration since smoking cessation, presenting smoking levels in a sequential way will more clearly show timing of smoking cessation and its relationship and trends with SUID. Additionally, keeping heavy and light smoker together for each smoking category will make it easier to evaluate dose response.

---

## [Decision Letter · Decision Letter 1]

24 Feb 2026

Maternal smoking behaviour during pregnancy and the association of Sudden Unexpected Infant Death (SUID): A retrospective cohort study of births in the United States from 2017-2021

PONE-D-25-50358R1

Dear Dr. Kiki Hudson,

We’re pleased to inform you that your manuscript has been judged scientifically suitable for publication and will be formally accepted for publication once it meets all outstanding technical requirements.

Kind regards,

Malshani Lakshika Pathirathna, PhD

Academic Editor

PLOS One

Additional Editor Comments (optional):

Reviewers' comments:

Reviewer's Responses to Questions

**Comments to the Author**

1. If the authors have adequately addressed your comments raised in a previous round of review and you feel that this manuscript is now acceptable for publication, you may indicate that here to bypass the “Comments to the Author” section, enter your conflict of interest statement in the “Confidential to Editor” section, and submit your "Accept" recommendation.

Reviewer #1: All comments have been addressed

Reviewer #2: All comments have been addressed

2. Is the manuscript technically sound, and do the data support the conclusions?

Reviewer #1: Yes

Reviewer #2: Yes

3. Has the statistical analysis been performed appropriately and rigorously? 

Reviewer #1: Yes

Reviewer #2: Yes

4. Have the authors made all data underlying the findings in their manuscript fully available?

Reviewer #1: Yes

Reviewer #2: Yes

5. Is the manuscript presented in an intelligible fashion and written in standard English?

Reviewer #1: Yes

Reviewer #2: Yes

6. Review Comments to the Author

Reviewer #1: (No Response)

Reviewer #2: In figure 2, the y-axis legend should read "Odds Ratio" not "Odds Ration".....I'm now typing to make the minimum character count requirment.

7. PLOS authors have the option to publish the peer review history of their article (what does this mean? ). If published, this will include your full peer review and any attached files.). If published, this will include your full peer review and any attached files.

**Do you want your identity to be public for this peer review?** For information about this choice, including consent withdrawal, please see our For information about this choice, including consent withdrawal, please see our Privacy Policy ..

Reviewer #1: No

Reviewer #2: **Yes:** William B WeeksWilliam B Weeks

---

## [Editor Report · Acceptance letter]

PONE-D-25-50358R1

PLOS One

Dear Dr. Hudson,

I'm pleased to inform you that your manuscript has been deemed suitable for publication in PLOS One. Congratulations! Your manuscript is now being handed over to our production team.

Kind regards,

on behalf of

Dr. Malshani Lakshika Pathirathna

Academic Editor

PLOS One